# Cell-lysis sensing drives biofilm formation in *Vibrio cholerae*

Jojo A. Prentice ●[1], Robert van de Weerd ●[1] & Andrew A. Bridges ●[1] ✉

Matrix-encapsulated communities of bacteria, called biofilms, are ubiquitous in the environment and are notoriously difficult to eliminate in clinical and industrial settings. Biofilm formation likely evolved as a mechanism to protect resident cells from environmental challenges, yet how bacteria undergo threat assessment to inform biofilm development remains unclear. Here we find that population-level cell lysis events induce the formation of biofilms by surviving *Vibrio cholerae* cells. Survivors detect threats by sensing a cellular component released through cell lysis, which we identify as norspermidine. Lysis sensing occurs via the MbaA receptor with genus-level specificity, and responsive biofilm cells are shielded from phage infection and attacks from other bacteria. Thus, our work uncovers a connection between bacterial lysis and biofilm formation that may be broadly conserved among microorganisms.

Surface-associated microbial communities, such as polymeric matrix-encapsulated biofilms, were among the earliest forms of life on Earth[1–6]. It has long been hypothesized that hostile environmental conditions selected for the evolution of biofilms[2,5,7]. Indeed, abundant experimental evidence has shown that the microbes found in biofilm communities are protected against a variety of threats, including diffusible antimicrobial compounds, phage, and attacks from other organisms, suggesting that biofilm formation is an ancient form of immunity[8–13]. The self-produced matrix that defines the biofilm structure protects resident microbes by limiting diffusion and physical contact between biofilm cells and threatening agents[9,10,13]. This property makes biofilms notoriously difficult to eradicate in clinical settings, where many pathogens form biofilms, and in industries, where biofilms are responsible for biofouling[14].

Given the benefits conferred to microbial cells that form encapsulated communities, it is no surprise that the biofilm lifestyle arose early and has persisted for billions of years. However, commitment to the biofilm lifestyle also presents challenges. Chief among them is that prolonged maturation of a biofilm community can lead to crowding and starvation[2]. As a consequence, many bacteria have evolved the ability to transition between the biofilm and free-swimming lifestyles and encode elaborate signaling mechanisms to regulate their collective states[15]. Yet, whether and how bacteria gauge the presence of lethal threats in their environments to drive biofilm formation remains unclear.

In this work, we set out to determine if bacteria harbor mechanisms by which they link threat detection to the collective formation of biofilms for protection. We find that upon exposure to lytic phages, the global pathogen *Vibrio cholerae* rapidly lyses, but then recovers, with surviving cells exhibiting robust biofilm formation. We find that the mechanism underlying this observation is a process we refer to as "lysis sensing," whereby surviving cells sense a signal released by the death of their kin. We identify the cell lysis signal as an abundant cytoplasmic polyamine, norspermidine, which is detected by an inner-membrane receptor, MbaA, that in turn drives biofilm formation. We determine that additional pathogenic organisms also exhibit multicellular community formation in response to lysis. Given the pervasiveness of lysis in the environment, we propose that lysis sensing is a threat-agnostic mechanism by which bacteria can gauge endangerment and respond by forming protective biofilms.

## Results

### Phage infection drives biofilm formation in *V. cholerae*

Bacteria face unrelenting attacks in their environments, reflected in estimates that phage lyse ~20% of the oceanic microbial biomass daily[16]. To assess whether bacteria alter their biofilm lifecycle in response to such threats, we began by studying the interaction between *V. cholerae* and the lytic phage S5[17]. In the absence of phage, *V. cholerae* cells exhibited sustained growth for ~20 h and remained planktonic over the course of the experiment (Fig. 1A and Supplementary Fig. 1a). By contrast, in the presence of phage (MOI = 10$^{-6}$), cells were initially planktonic, but after ~3–5 h of cell death due to lysis, surviving cells formed verticalized, multicellular structures, which

[1]Department of Biological Sciences, Carnegie Mellon University, Pittsburgh, PA, USA. ✉e-mail: bridges@cmu.edu

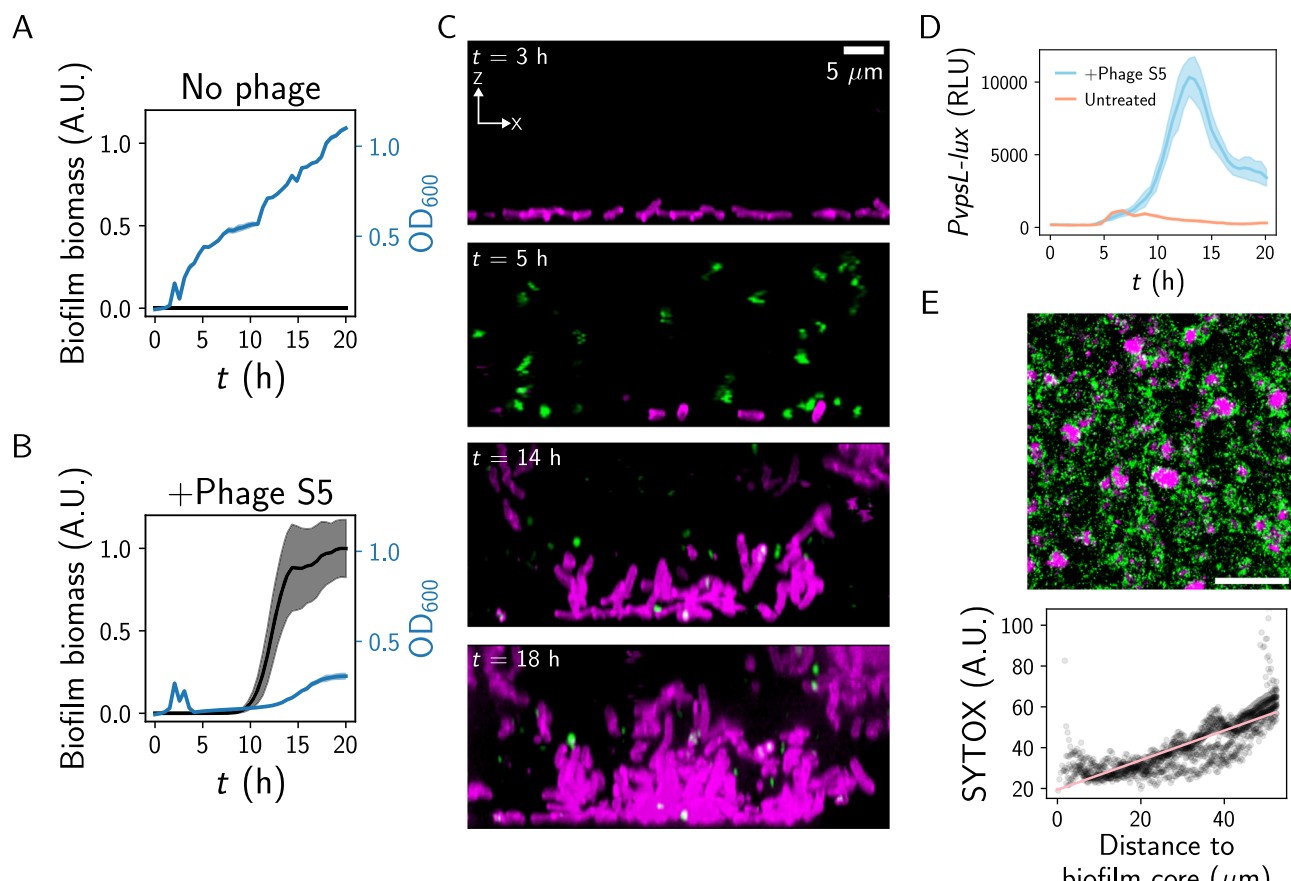

**Fig. 1 | *V. cholerae* forms biofilms in response to a lytic phage. A** OD$_{600}$ growth curve and biofilm biomass curve as measured by brightfield microscopy for wild-type *V. cholerae* grown in the absence of phage. **B** As in (**A**) for wild-type *V. cholerae* grown with phage S5. **C** Confocal images for the indicated timepoints of *V. cholerae* grown with phage S5. Magenta represents live cells expressing a constitutive reporter and green represents dead cells stained with SYTOX. Axes and scale bar are as indicated in the top panel. **D** *PvpsL-lux* output over time from wild-type *V. cholerae* grown in the indicated conditions. **E** Top: representative time projection of 20x confocal images of wild-type *V. cholerae* grown with phage S5. Colors are as in (**C**). Scale bar is 100 μm. Bottom: quantification of SYTOX green (dead-cell) signal as a function of distance to the biofilm core, based on 20× confocal images of wild-type *V. cholerae* grown with phage S5. Pink line represents a best-fit line for the data ($R^2$ = 0.687). **A**, **B** Biofilm biomass data are normalized to the peak biofilm biomass of the phage-treated condition. **A**, **B**, **D** Data represent averages of $n$ = 3, 6, and 6 biological replicates, respectively, and shading represents standard deviations. A.U. arbitrary units, RLU relative-light units.

grew for ~12 h and were sustained thereafter (Fig. 1B, C and Supplementary Movie 1). Both biofilm formation and growth recovery were observed at a range of MOIs (up to MOI ~0.1) in response to infection (Supplementary Fig. 1b). We tested whether these multicellular structures were canonical biofilms by assessing the expression of an exopolysaccharide biosynthesis gene (*vpsL*) via an established luciferase promoter fusion[18,19]. *PvpsL-lux* output increased ~11-fold in the presence of phage (Fig. 1D), confirming that growth in the presence of lytic phage drives biofilm gene expression in *V. cholerae*. Of note, when we treated *V. cholerae* with the lytic phage N4, which exhibits a larger burst size (~50 particles/cell compared to ~10 particles/cell for S5; Supplementary Fig. 1c), we observed a similar increase in biofilm formation and elevated *PvpsL-lux* output relative to untreated cultures, showing that the biofilm response to lytic phage is not unique to a particular phage (Supplementary Fig. 1c)[17]. Finally, by analyzing low-magnification images of dozens of biofilms that formed in response to phage S5, we observed that dead cells were more prevalent toward the edges of the biofilms compared to the biofilm cores (Fig. 1E). This suggests that the biofilm structure serves as a shield against phage infection, as has been demonstrated in previous studies[10,20]. Together, these results show that during their interactions with a lytic phage, *V. cholerae* cells protect themselves from infection by committing to the formation of a matrix-embedded multicellular community.

## *V. cholerae* drives biofilm formation in response to a cellular component released through lysis

We sought to identify the mechanism through which *V. cholerae* controls biofilm formation in response to lytic phages. We considered three possibilities: (1) phage killing selects for cells that exhibit elevated *vps* expression, (2) *V. cholerae* cells sense phage components during infection and form biofilms in response, or (3) lysis releases a cytoplasmic signal, and living cells respond to this cue by committing to the biofilm state. We reasoned that one could distinguish between these possibilities by exposing cells to mechanically produced cell lysate lacking any phage component. To this end, we assessed the biofilm dynamics of *V. cholerae* grown in the presence or absence of its own lysate. In minimal medium, untreated cells in static culture grew into biofilms for ten hours, at which point they initiated biofilm dispersal, resulting in a planktonic culture by fifteen hours (Fig. 2A, Supplementary Movie 2, and Supplementary Fig. 2). By contrast, in the presence of lysate generated by mechanical disruption of overnight cultures, biofilm biomass increased approximately threefold, and we observed attenuated biofilm dispersal (Fig. 2A and Supplementary Movie 2), indicating that *V. cholerae* cells commit to the biofilm state in the presence of *V. cholerae* lysate. Reconstitution of the biofilm response to phage with mechanically produced lysate suggests that no phage component is required for the response observed in Fig. 1, and that cells commit to the biofilm state in response to a cellular factor

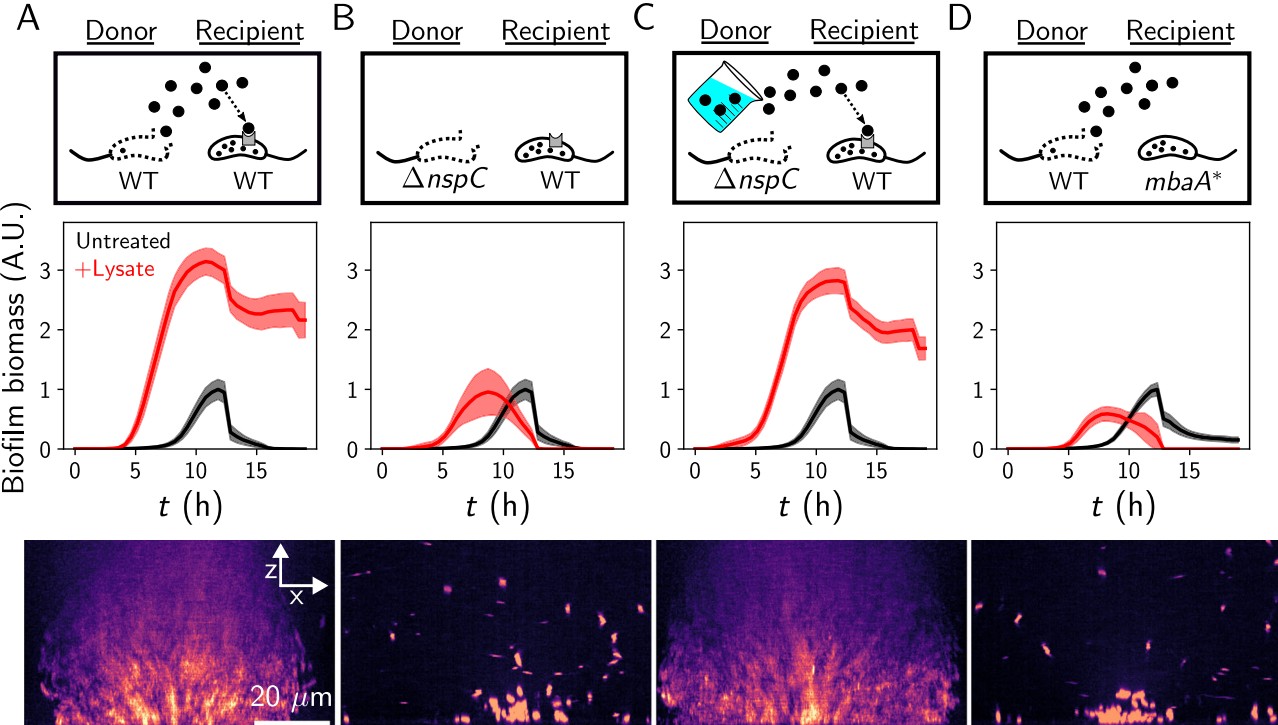

**Fig. 2 | Self-derived lysate drives biofilm formation via norspermidine signaling. A** Top panel: Schematic of cell-lysate donor and recipient strains, with norspermidine denoted by black circles. Middle panel: Quantified biofilm lifecycles from brightfield microscopy for the recipient strain grown with or without mechanically produced donor lysate, as indicated. Bottom panel: representative XZ confocal projection of the recipient strain treated with donor lysate after 16 h. Cells were stained with MM4-64, displayed with the mpl-magma look-up table. **B** As in (**A**) where the Δ*nspC* strain serves as the lysate donor and wild-type is the recipient.

**C** As in (**A**) where the Δ*nspC* donor lysate was supplemented with 100 μM of synthetic norspermidine and wild-type is the recipient. **D** As in (**A**) where the wild-type strain serves as the lysate donor and the *mbaA** strain is the recipient. Data are normalized to the peak biofilm biomass of the untreated wild-type. The wild-type untreated control in (**A**) served as the control for (**B**, **C**), as experiments were carried out simultaneously. Data represent averages of n = 3 biological replicates, n = 3 technical replicates of the recipient, and n = 2 biological replicates of the lysate donor. Shading represents standard deviations. A.U. arbitrary units.

released by lysis, which we hereafter refer to as lysis sensing. Given the pervasiveness of lysis in the environment and the diversity of lytic threats, we reason that lysis sensing could serve as a threat-agnostic mechanism that enables *V. cholerae* cells to detect a challenge and respond by collectively protecting themselves.

**Norspermidine is the lysis signal**

We next set out to identify the *V. cholerae* lysis-sensing signal. We considered the polyamine norspermidine as a candidate, because it is abundant in the *V. cholerae* cytoplasm, is not secreted, and, to our knowledge, is the only self-produced small molecule that has been shown to drive the biofilm state in *V. cholerae*[21–23]. To test our hypothesis, we treated wild-type cells with lysate derived from a Δ*nspC* mutant, which does not encode the carboxynorspermidine decarboxylase required for norspermidine biosynthesis (Supplementary Fig. 3)[24]. In response to the norspermidine deficient lysate, wild-type cells did not exhibit elevated biofilm biomass relative to untreated cells (Fig. 2B and Supplementary Movie 2). However, when we administered lysate from the Δ*nspC* strain spiked with exogenous norspermidine, we observed robust biofilm formation and attenuated dispersal, akin to the response to wild-type lysate (Fig. 2C). Thus, norspermidine is necessary and sufficient for the biofilm response to lysate in *V. cholerae*.

Previous studies have demonstrated that norspermidine regulation of biofilm formation occurs through an inner-membrane receptor called MbaA. Briefly, when extracellular norspermidine is abundant, it is detected by the periplasmic protein NspS, which associates with MbaA, driving MbaA to synthesize the conserved biofilm-promoting second messenger c-di-GMP (Supplementary Fig. 3)[21–25]. Thus, we

predicted that functional MbaA should be required for lysis sensing. To test this hypothesis, we assessed lysis sensing in a strain containing a norspermidine-unresponsive *mbaA* allele that harbors a mutated active site (hereafter *mbaA**)[22]. In response to wild-type lysate, *mbaA** mutant cells did not alter their biofilm lifecycle compared to untreated *mbaA** cells (Fig. 2D, Supplementary Movie 2, and Supplementary Fig. 2). Together, these results indicate that *V. cholerae* cells commit to the biofilm state in response to lysate, and that the response is mediated by norspermidine signaling through the MbaA receptor.

For a lysis-sensing system to confer population-level protection from lytic threats, we reasoned that it would be important for bacteria to respond to a minority of cells lysing in the population. By treating wild-type *V. cholerae* cells with a range of lysate concentrations and calibrating the lysate to known concentrations of norspermidine, we found that lysis of a dense overnight culture (10^9 CFUs/mL) resulted in the release of ~30–50 μM norspermidine, and that less than 1% of this lysate was required to elevate $P_{vpsL}$-*lux* output 25-fold. Thus, lysis of 10^6–10^7 cells/mL (corresponding to an OD_60 ~0.01–0.1) was sufficient to bias *V. cholerae* cells toward the biofilm state (Supplementary Fig. 4a–c), consistent with our original finding of phage-induced biofilm formation in Fig. 1. These results suggest that in an exponentially growing population, lysis of a subpopulation of *V. cholerae* cells releases sufficient norspermidine to drive biofilm formation via the MbaA receptor.

**Lysis sensing mediates the phage-driven response and protects cells from lytic threats**

To evaluate whether norspermidine-mediated lysis sensing explains the biofilm response to lytic phage observed in Fig. 1, we measured

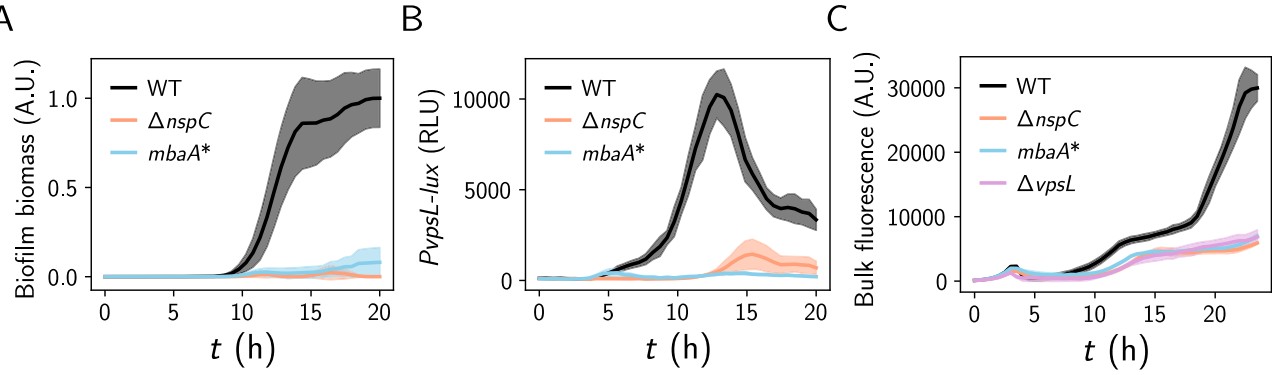

**Fig. 3 | Norspermidine-mediated lysis sensing enhances *V. cholerae* recovery from phage infection by driving biofilm formation. A** Biofilm biomass over time for the indicated strains grown with phage S5. Data are normalized to the peak biofilm biomass of wild-type with phage. **B** *PvpsL-lux* output over time from the indicated strains grown with phage S5, expressed as relative-light units (RLU = luminescence/OD$_{600}$). **C** Bulk culture fluorescence emitted from the constitutively expressed reporter *Ptac-dL5* over time from the indicated strains grown with phage S5. Data represent averages of $n = 6$ biological replicates of each *V. cholerae* strain and shading represents standard deviations. A.U. arbitrary units, RLU relative-light units.

biofilm formation in the presence of phage for the lysis-sensing deficient mutants, $\Delta nspC$ and $mbaA*$. In contrast to the wild-type strain, neither mutant strain formed substantial biofilms in the presence of phage, showing that norspermidine signaling through the MbaA pathway is required for the biofilm response (Fig. 3A and Supplementary Movie 3). To validate our findings, we evaluated biofilm gene expression using *PvpsL-lux*. Consistent with our biofilm measurements, *PvpsL-lux* output from the wild-type strain was approximately sevenfold greater than that of the $\Delta nspC$ mutant and was ~24-fold greater than that of the $mbaA*$ mutant in the presence of phage (Fig. 3B). These results demonstrate that lysis sensing, and not selection of cells exhibiting elevated *vps* gene expression, accounts for the elevation in biofilm formation in the presence of phage.

We note that the $\Delta nspC$ and $mbaA*$ mutants did appear to form aggregate-like biofilms in the presence of phage that were not observed in the wild-type strain and, due to their distinct optical properties, were not segmented by our image analysis pipeline developed for detecting canonical VPS-dependent biofilms (Supplementary Fig. 5 and Supplementary Movie 3). Of note, these aggregate biofilms were also observed in a $\Delta vpsL$ strain, which lacks a critical polysaccharide matrix biosynthesis enzyme required for canonical VPS-dependent biofilm formation (Supplementary Fig. 5). We speculate that the aggregate biofilms observed in the aforementioned mutant strains could result from cellular polymers released by lysis that induce aggregation by a depletion mechanism, as previously observed for other bacterial species[26].

To examine whether biofilm formation driven by lysis sensing is advantageous during phage challenge, we monitored growth curves for the wild-type, $\Delta nspC$ and $mbaA*$ strains using a constitutive fluorescent reporter. We found that population-level recovery after lysis was elevated in the wild-type strain (~threefold higher at 24 h) compared to the lysis-sensing deficient strains (Fig. 3C), demonstrating that norspermidine-mediated lysis sensing conveys a strong survival advantage in the presence of phage. Moreover, the $\Delta vpsL$ mutant, which is incapable of forming biofilms, exhibited an identical recovery curve to the lysis-sensing deficient mutants (Fig. 3C), showing that protection is mediated through biofilm formation. Cumulatively, these results show that lysis sensing endows *V. cholerae* with the ability to sense and respond to lytic threats by forming biofilms, which in turn protects cells during recovery.

Although this study has primarily focused on the relationship between norspermidine signaling and protection from phage-driven lysis, lysis sensing could in principle serve a similar benefit in contexts in which other lytic agents are present. We wondered whether pre-exposure to norspermidine would protect *V. cholerae* cells against Type-

VI secretion system attacks, a prevalent threat to surface-associated bacterial cells in the environment[27]. To this end, we treated a population of *V. cholerae* cells with norspermidine and examined the effects on survival against Type-VI secretion system attacks from *Acinetobacter baylyi* ADP1[13]. We found that norspermidine-treated wild-type cultures exhibited a >200-fold survival advantage compared to untreated cultures after exposure to *A. baylyi*. In contrast, norspermidine-treatment of $mbaA*$ cultures, which are unable to respond to the lysis signal, were not protected relative to untreated $mbaA*$ cultures (Supplementary Fig. 6). Notably, pre-exposure to the lysis signal was required for a survival advantage as we observed no survival difference between the untreated wild-type and $mbaA*$ strains in this standard T6SS competition assay. We reason that due to the short timescale of incubation (2 h), lysis sensing occurring concomitant with exposure to high number of *A. baylyi* cells does not provide enough time for biofilm-mediated protection to be enacted before *V. cholerae* is overwhelmed by attacks. These results suggest that the lysis signaling mechanism that allows cells to collectively respond to phage-mediated killing may allow them to respond similarly to other lytic threats.

### *V. cholerae* responds to cell lysis with genus-level specificity
Given the role of lysis sensing in *V. cholerae*-phage population dynamics, we were motivated to probe the evolutionary origins of the lysis-sensing pathway. Through bioinformatic analysis, we found that the critical norspermidine-producing enzyme, a fusion of L-2,4-diaminobutyrate aminotransferase (DABA AT) and L-2,4-diaminobutyrate decarboxylase (DABA DC), is present almost exclusively in members of the Vibrionaceae family, and predominantly within the *Vibrio* genus (Supplementary Table 1)[24]. Based on this observation, we wondered whether lysis sensing by *V. cholerae* occurs with kin lysate specificity. Pervasive lytic threats (e.g., phage) often target prey with high specificity—down to the level of single strains or isolates, and thus, threat assessment is likely most relevant if information about the threat is encoded in a kin-specific lysis signal. To test the specificity of lysis sensing, we prepared lysates from a set of related bacterial species that are predicted to produce norspermidine (*Vibrio anguillarum*, *Vibrio campbellii*, *Vibrio natriegens*, *Vibrio parahaemolyticus*, and *Vibrio vulnificus*; Fig. 4A, B), as well as select bacterial species that are more distantly related and are not predicted to produce norspermidine (*Aliivibrio fischeri*, *Escherichia coli*, and *Pseudomonas aeruginosa*; Fig. 4A, B). Consistent with our hypothesis, only lysates produced from bacteria encoding DABA AT/DC drove *V. cholerae* cells to commit to the biofilm state, whereas, intriguingly, lysates from bacteria that do not produce norspermidine repressed *V. cholerae* biofilm formation altogether (Fig. 4B and Supplementary Movie 4).

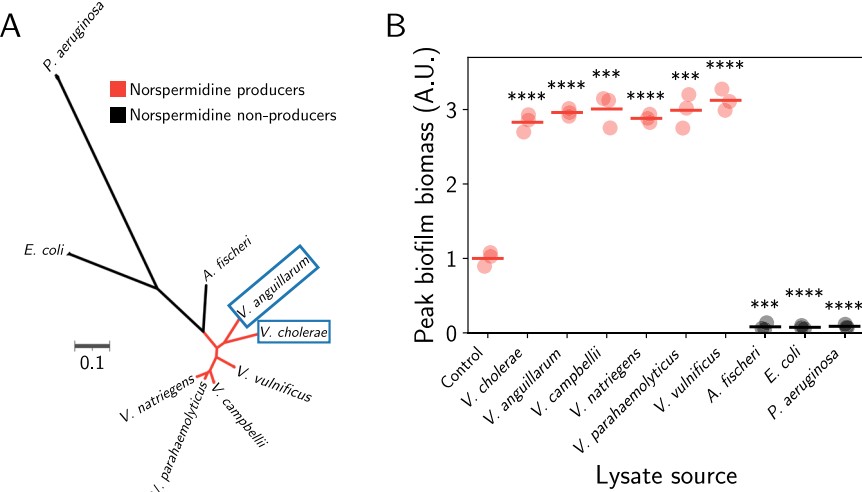

**Fig. 4 | *V. cholerae* commits to the biofilm state in response to lysates from members of the *Vibrio* genus. A** Unrooted maximum-likelihood phylogenetic tree of the organisms studied in this work, as determined by core gene alignments. Scale bar represents substitutions per amino acid site. Colors denote predicted norspermidine producers and non-producers, as indicated. Blue boxed species encode MbaA. **B** Peak biofilm biomass of *V. cholerae* in response to mechanically produced lysates from the indicated organisms. Data represent $n = 3$ biological replicates of the recipient strain and are normalized to the mean peak biofilm biomass of the control (no lysate). Points represent individual replicates and crossbars represent means. Colors are as in (**A**). Statistical significance is indicated based on unpaired, two-sided *t* tests relative to the untreated control. $P = 3.13 \times 10^{-5}$, $6.07 \times 10^{-6}$, $1.32 \times 10^{-4}$, $7.65 \times 10^{-6}$, $1.49 \times 10^{-4}$, $2.81 \times 10^{-5}$, $1.15 \times 10^{-4}$, $7.96 \times 10^{-5}$, $8.62 \times 10^{-5}$ for the indicated pairs. ****$P \leq 0.0001$. ***$0.0001 < P \leq 0.001$. A.U. arbitrary units.

We wondered whether *V. cholerae* cells aborted their biofilm lifecycle in response to lysates from species that do not produce norspermidine because they sensed spermidine. Spermidine is a polyamine that is broadly produced outside of the *Vibrio* genus and is known to drive *V. cholerae* cells to commit to the planktonic state by switching MbaA activity from c-di-GMP production to c-di-GMP degradation (Supplementary Fig. 3)[22,23,28]. To test this hypothesis, we prepared lysate from an *E. coli* strain that lacks one of the essential enzymes for spermidine biosynthesis (*speE::kanR*) and observed its effect on *V. cholerae* biofilm formation. Contrary to our hypothesis, *V. cholerae* cells repressed biofilm formation in response to lysate from this strain (Supplementary Fig. 7), suggesting that upon lysis of species outside of the *Vibrio* genus, *V. cholerae* cells sense an additional unknown signal and repress biofilm formation in response.

### Diverse *Vibrios* form multicellular communities in response to lysate

We next considered the phylogenetic distribution of the norspermidine sensing machinery. We found that only a subset of *Vibrios*, belonging to the clade of *V. cholerae* and *V. anguillarum*, encode putative homologs of MbaA (Fig. 4A and Supplementary Table 2). We conclude that only members of this clade are likely to respond to norspermidine. To validate this conclusion experimentally, we treated *V. anguillarum*, as well as *V. parahaemolyticus* and *V. vulnificus*, which do not encode a clear MbaA homolog, with norspermidine and examined the effects on multicellularity by brightfield and confocal fluorescence microscopy. Consistent with the bioinformatic prediction, only *V. anguillarum* cells drove multicellular community formation in response to norspermidine (Fig. 5A–C and Supplementary Movie 5). Moreover, as expected, *V. anguillarum* developed robust surface-associated communities in response to its own lysate and to wild-type *V. cholerae* lysate, and the response was substantially attenuated when we used lysate from the *V. cholerae* Δ*nspC* strain (Fig. 5A, Supplementary Movie 5, and Supplementary Fig. 8). Notably, however, *V. anguillarum* exhibited greater biofilm formation in the presence of its own lysate than in the presence of synthetic norspermidine (Fig. 5A), suggesting that other cellular components may synergize with norspermidine to drive lysis sensing in this bacterium.

We note that a previous study observed that *V. anguillarum* formed biofilms in response to phage, but the mechanism underpinning the response was not identified[29]. Our results show that the norspermidine-dependent lysis-sensing mechanism is functionally conserved in a subset of *Vibrio* species, and that lysis sensing could explain the previous observation of phage-induced biofilm formation in *V. anguillarum*.

We wondered whether the norspermidine-unresponsive *Vibrios* harbored a distinct mechanism by which they could sense and respond to cell lysis. To test this possibility, we grew *V. parahaemolyticus* and *V. vulnificus* in the presence or absence of their own lysates. In the absence of lysate, *V. parahaemolyticus* cells grew as a planktonic culture and *V. vulnificus* exhibited modest multicellular community formation (Fig. 5B, C, Supplementary Movie 5, and Supplementary Fig. 8). By contrast, the addition of cell lysate drove both organisms to form millimeter-scale communities (Fig. 5B, C and Supplementary Movie 5). We observed similar structures upon addition of wild-type or Δ*nspC V. cholerae* lysates, confirming that the signal used by these organisms to sense lysis is made by diverse *Vibrios* but is not norspermidine (Fig. 5B, C and Supplementary Movie 5). Together, these results show that lysis sensing via norspermidine is constrained to a clade of *Vibrios* that includes *V. cholerae* and *V. anguillarum*, but not *V. parahaemolyticus* and *V. vulnificus*. However, the latter two form multicellular communities in response to lysis, suggesting that the connection between lysis sensing and multicellularity extends beyond the norspermidine signaling pathway in the *V. cholerae* clade and may have evolved multiple times.

## Discussion

Bacteria have evolved elaborate signaling pathways that allow them to link their behaviors to the demands of their environments; molecular circuits that respond to metabolic cues, host signals, and quorum-sensing autoinducers are widespread in the biosphere and are known to regulate multicellularity in diverse microorganisms. Here, we have demonstrated that *V. cholerae* detects threats by kin lysis sensing, which drives multicellularity and protects cells from threats. Unlike most of the known bacterial innate immune defenses, which cells activate upon infection by phage, the form of lysis sensing we have

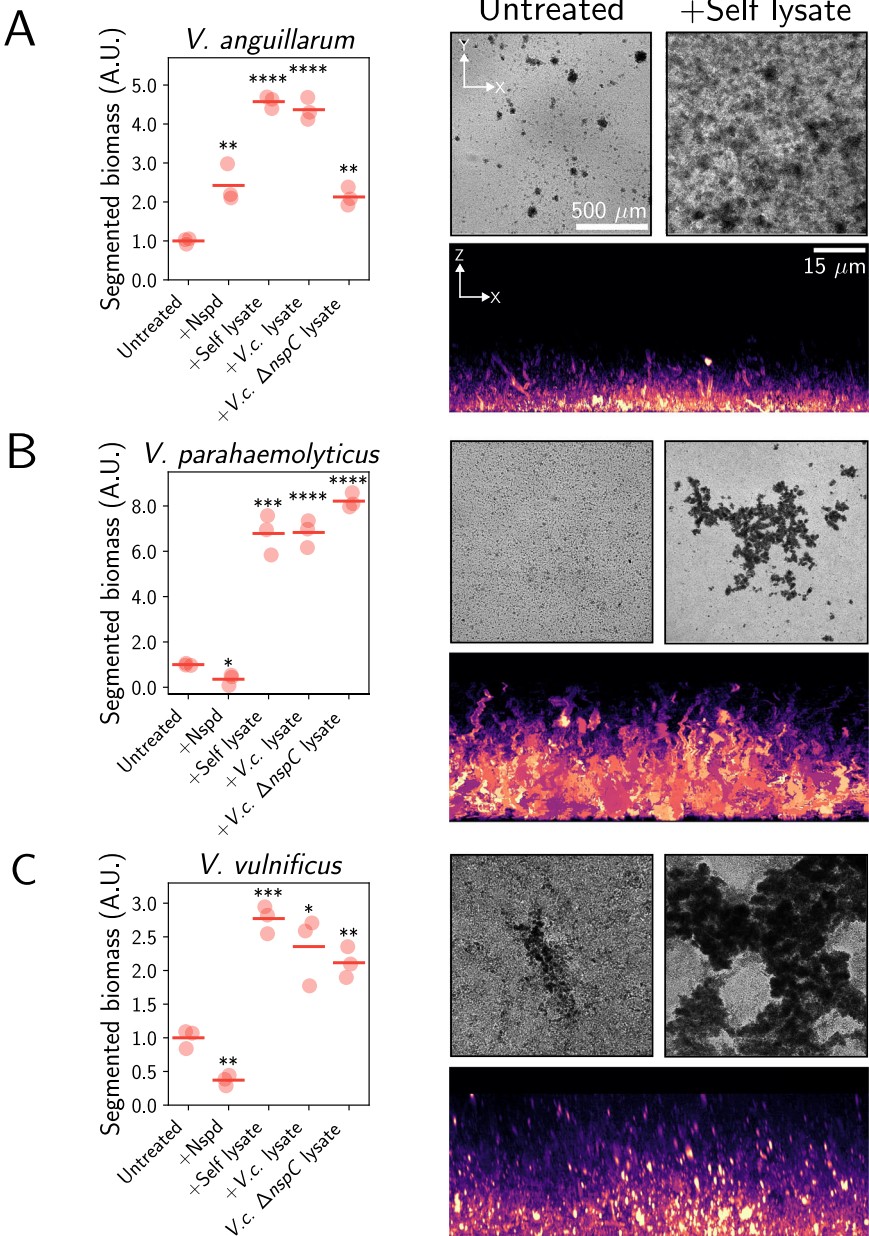

**Fig. 5 | Lysis sensing drives multicellularity in diverse *Vibrios* via norspermidine-dependent and -independent pathways. A** Left panel: peak segmented biomass of *V. anguillarum* in response to the indicated growth conditions. Data represent $n = 3$ biological replicates of the recipient strain and are normalized to the mean peak segmented biomass of the untreated control. Points represent individual replicates, and crossbars represent means. Statistical significance is indicated based on unpaired, two-sided $t$ tests relative to the untreated control. $P = 7.21 \times 10^{-3}$, $3.49 \times 10^{-6}$, $3.87 \times 10^{-5}$, and $1.32 \times 10^{-3}$ for the indicated pairs. Right panels, top: representative brightfield images of *V. anguillarum* without lysate (left) and treated with mechanically produced *V. anguillarum* lysate (right) after 16 h of growth. Right panels, bottom: Representative XZ projection of a confocal z-stack of *V. anguillarum* grown with its own lysate for 16 h. **B** As in (**A**) for *V. parahaemolyticus*. $P = 1.02 \times 10^{-2}$, $3.39 \times 10^{-4}$, $7.72 \times 10^{-5}$, and $2.99 \times 10^{-6}$ for the indicated pairs. Images were acquired after 12 h of growth. **C** As in (**A**) for *V. vulnificus*. $P = 2.42 \times 10^{-3}$, $2.43 \times 10^{-4}$, $1.12 \times 10^{-2}$, and $1.97 \times 10^{-3}$ for the indicated pairs. In all cases, cells were stained with the lipophilic dye MM4-64, displayed with the mpl-magma look-up table in ImageJ. Scale bars are as indicated in the *V. anguillarum* images. A.U. arbitrary units, Nspd norspermidine, *V.c. Vibrio cholerae*. ****$P \le 0.0001$. ***$0.0001 < P \le 0.001$. **$0.001 < P \le 0.01$. *$0.01 < P \le 0.05$.

described here directly prevents encounters with generic lytic threats[30]. We reason that by temporarily preventing confrontation, bacterial cells gain a reprieve to enact adaptive protective mechanisms. Our results suggest that despite the idiosyncrasies of species-specific responses, the concept of committing to a biofilm state in response to kin-cell lysis may be widespread in microorganisms. Future work will be required to determine whether lysis-sensing control of biofilm formation extends beyond the particular species and strains employed in this study.

We found that norspermidine signaling through the MbaA receptor is necessary and sufficient for *V. cholerae* cells to collectively commit to the biofilm state in response to lysis, and that this commitment protects cells from phage infection and Type-VI secretion system killing. Previous studies of *P. aeruginosa* have demonstrated that exposure to polyamine danger signals present in lysate, in conjunction with the presence of linear DNA, limits phage replication[31,32]. It remains possible that a biofilm response also contributes to the *P. aeruginosa* protection mechanism. The polyamine signaling systems in *P. aeruginosa* and *V. cholerae*

highlight a potential conservation of polyamine-mediated lysis sensing. We reason that polyamines are prime candidates for identifying novel lysis-sensing systems, given their abundances in the bacterial cytoplasm and their ancient evolutionary origins[33].

The biophysical characteristics of threats and the properties of lysis-sensing circuits (e.g., sensitivity) are likely critical for the effectiveness of lysis sensing in protecting bacterial populations. Regarding the properties of threats, we reason that the amount of the threatening agent, its diffusivity, and its propagation rate are crucial parameters. A population of bacteria presumably cannot protect themselves against a threat that diffuses rapidly and affects all members of a population simultaneously (e.g., antibiotics, or a very high concentration of phage) using lysis sensing, which depends on the comparatively longer timescales of lysate accumulation and the response to a lysis signaling molecule. On the other hand, threats that propagate through a population over time and diffuse slower than lysis signals, such as dilute phage and bacterial or protozoan predators, are likely candidates that could drive the evolution of lysis sensing.

Bacteria often encounter multiple environmental inputs simultaneously that can either synergize or interfere with one another via signal integration networks. Investigating how cells integrate lysis sensing with other signaling inputs could reveal the ecological settings in which lysis sensing is most relevant. Previous results indicated a relationship between the lysis-sensing receptor, MbaA, and quorum sensing. Quorum sensing is the process by which bacteria secrete and detect small molecule signals, allowing them to measure the cell density and relatedness of their community[34]. Bacteria including *V. cholerae* have been shown to upregulate other phage protection mechanisms at high cell densities via quorum sensing, presumably to protect themselves when the population is most vulnerable to phage propagation, i.e., when cell numbers are large[35]. We previously showed that quorum sensing, which under normal conditions represses biofilm formation in *V. cholerae*, amplifies biofilm formation in the presence of the norspermidine signal via upregulation of the MbaA receptor[36]. Thus, *V. cholerae* cells are more sensitive to lysis signals when the potential for phage propagation is highest. Based on these findings, and their functional logic, we speculate that a regulatory link between lysis sensing and quorum sensing may be common in bacteria.

To date, studies have largely conceptualized the biofilm lifecycle as the execution of a plan, akin to embryogenesis[37,38]. How bacterial populations alter their developmental trajectories according to the information they garner from interspecies and interkingdom interactions, which are prevalent in the environment, remains a ripe area for investigation. The current work presents a molecular and population-scale perspective on one class of such interactions, namely that of bacteria and lytic phage (host-parasite). Continued mechanistic studies of other classes of interactions (e.g., competitive or symbiotic) will help bridge the molecular and ecological scales of bacterial biofilm dynamics. We propose that the molecular architectures underlying these interactions may be key drivers of bacterial lifestyle decisions in natural contexts.

## Methods

### Bacterial growth, antibiotics, and cloning

All strains used in this work are reported in Supplementary Table 3. *V. cholerae* strains, *A. baylyi* ADP1, *P. aeruginosa*, and *E. coli* were propagated on lysogeny broth (LB) plates supplemented with 1.5% agar or in liquid LB with shaking at 30 °C. *A. fischeri* and all other *Vibrio* species were propagated on lysogeny broth (LB) plates supplemented with 1.5% agar and 2% NaCl or in liquid LB + 2% NaCl with shaking at 30 °C. For biofilm experiments *V. cholerae* strains were grown in M9 minimal medium containing dextrose and casamino acids (1× M9 salts, 100 μM CaCl₂, 2 mM MgSO₄, 0.5% dextrose, 0.5% casamino acids), whereas other *Vibrio* species were grown in the same medium with an additional 2% NaCl. For all assays in which *V. cholerae* was grown with

phage, bacteria and phage were propagated together in LB containing 0.5% dextrose and 10 mM CaCl₂ (referred to as phage medium) to permit infection. When necessary, antimicrobials were supplied at the following concentrations: polymyxin B, 25 μg/mL; kanamycin, 50 μg/mL; spectinomycin, 200 μg/mL. Unless otherwise indicated, norspermidine (Millipore Sigma, I1006-100G-A) was added at 100 μM at the start of each assay. Modifications to the *V. cholerae* genome were generated by replacing genomic DNA with linear DNA introduced by natural transformation, as described previously[39–41]. PCR and Sanger sequencing (Azenta) were used to verify genetic alterations. Oligonucleotides and synthetic linear DNA g-blocks were ordered from IDT and are reported in Supplementary Table 4.

### Microscopy

General biofilm growth procedures, which pertain to Figs. 2, 4B, and 5, were carried out similarly to previous reports[19,41]. Briefly, samples were statically grown in 96-well polystyrene microtiter dishes (Corning). Brightfield timelapse microscopy images were acquired at 30-min intervals at 30 °C on an Agilent Biotek Cytation 1 imaging plate reader using either a 10× air objective (Olympus Plan Fluorite, NA 0.3) (Figs. 2 and 4) or a 4× air objective (Olympus Plan Fluorite, NA 0.13) (Fig. 5) driven by Biotek Gen5 (Version 3.12) software.

Spinning disc confocal images were acquired using a motorized Nikon Ti-2E stand outfitted with a CREST X-Light V3 spinning disk unit, a back-thinned sCMOS camera (Hamamatsu Orca Fusion BT) and a 100x silicone immersion objective (Nikon Plan Apochromat, NA 1.35) or a 20x air objective (Nikon Plan Apochromat, NA 0.75) (Fig. 1E) driven by Nikon Elements software (Version 5.42.02). Samples were incubated at 25 °C using a stage-top incubator (Oko Labs) and the source of illumination was an LDI-7 Laser Diode Illuminator (89-North). For the timelapse in Fig. 1C, E, live cells were labeled with 1 μM of the fluorogen malachite-green coupled to diethylene glycol diamine (MG-2p), which fluoresces upon stable binding to the fluorogen-activating protein dL5, which our strains produced[42–44]. To achieve robust dL5 expression, strains harbored *Ptac-ssMBP-dl5* chromosomally integrated at the *vc1807* neutral locus. The *tac* promoter drove strong constitutive expression and the secretion signal of maltose binding protein (ssMBP) was used to target dL5 to the periplasm. dL5-MG-2p fluorescence was excited at 640 nm. For single-timepoint confocal images (Figs. 2A–D and 5), cells were stained with 10 μM of the lipophilic dye MM4-64 (AAT Bioquest) and were imaged with an excitation wavelength of 561 nm. Dead cells were labeled with 1 μM of the membrane-impermeable DNA stain SYTOX Green (ThermoFisher) and imaged with an excitation wavelength of 488 nm. Samples were grown in 96-well glass-bottom microtiter dishes (Mattek).

### Image analysis

Biofilm biomass quantifications were performed similarly to previous studies[19,41]. Specifically, multicellular structures, when imaged by brightfield microscopy, attenuate light to a greater extent than planktonic cells. Our approach, therefore, was to segment these multicellular structures based on pixel intensity thresholding and subsequently to measure the total amount of light attenuated by all communities in the field of view. For segmentation of *V. cholerae*, *V. anguillarum*, and *V. parahaemolyticus* multicellular structures (which generate both in- and out-of-focus light attenuation), pixel intensities were inverted, the local contrast was normalized, the image was blurred with a Gaussian filter, and a constant threshold was applied to generate a mask. A median filter was then applied to remove noise from the mask. For *V. vulnificus* segmentation, a unique protocol was required, as this organism produces large multicellular structures whose intensities are suppressed by local contrast normalization, and minimal out-of-focus light attenuation is generated. In this case, a Variational Bayesian estimate of a Gaussian mixture was fitted to the intensity data, the smallest mean from the mixture was subtracted, and

a constant threshold was applied to generate a mask. Subsequently, morphological operations were applied, and the mask was blurred with a median filter. For all organisms, after segmentation, the mask was applied to the inverted image, and the intensities from the masked biofilms were summed to yield the biofilm biomass for that image. The peak biofilm biomass (Figs. 4B and 5) corresponds to the maximum biofilm biomass value within a given timelapse for each replicate. All image analyses were performed in Python 3.

For quantification of biofilm biomass in experiments with phage (Figs. 1A, B and 3A), we distinguished biofilms from the aggregate-type structures that the $\Delta vpsL$ biofilm mutant formed by placing an additional size constraint on the connected components in the mask. Any components that formed structures with an area greater than $0.026\,mm^2$ in the mask were filtered out, leaving only clonal biofilm structures.

To quantify SYTOX signal as a function of distance from the biofilm core (Fig. 1E), a rolling-ball background subtraction was applied to single transverse plane 20× spinning disc confocal images of both channels (SYTOX and constitutive) for all timepoints. A median filter was applied to remove salt-and-pepper noise, and biofilm masks were generated by applying a constant intensity threshold to the constitutive channel, followed by binary morphological operations and median filter smoothing. Centers of biofilms were determined by locating contours in the biofilm mask, applying a distance transform, and finding maxima of the distance transform inside each contour. SYTOX signal was then quantified as a function of distance from the biofilm centers for all timepoints. Figures and original cartoons were assembled in Inkscape software (v1.2.2).

### Phage procedures

*V. cholerae* phages S5 and N4 were obtained from ATCC (51352-B2 and 51352-B1, respectively). High titer phage stocks were produced by confluent plaquing of a wild-type *V. cholerae* lawn on top agar (LB 0.6% agar), followed by resuspension in 5 mL SM Buffer (100 mM NaCl, 8 mM $MgSO_4$, 50 mM Tris-Cl pH 7.5) and filtration through a 0.45-μM syringe filter (Pall Corporation, aerodisc). Filtration of phage preparations through 0.2 μm or 0.45 μm syringe filters resulted in similar phage preparations. Plaque-forming units of stocks were measured by tenfold serial dilutions of phage on wild-type *V. cholerae* lawns. One-step growth curves were performed at 32 °C to discern the phage N4 burst size, as described previously[45].

For phage infection experiments (Figs. 1 and 3), single colonies of the relevant *V. cholerae* strains were grown at 30 °C in phage medium to mid-log ($OD_{600} = 0.3$–$0.5$), followed by back-dilution in phage medium to $OD_{600} = 0.1$ in the presence of phage S5 at MOI = $\sim10^{-6}$ (or higher MOI's; Supplementary Fig. 1b). Cultures were subsequently grown statically at 25 °C in 96-well plates and either brightfield or spinning disc confocal microscopy was performed as described above. Under the same conditions, the promoter activity of the biofilm gene *vpsL* was monitored using an established luciferase reporter (*PvpsL-lux*) on an Agilent Biotek Cytation 1 Plate Reader via filtered luminescence (Figs. 1D and 3B)[18,19]. To measure recovery from phage killing (Fig. 3C), cultures were grown with phage as above. All strains harbored *Ptac-ssMBP-dL5*, as described above (Microscopy methods), and bulk constitutive fluorescence signal from each culture was measured at 30-min intervals in the presence of 1 μM MG-2p on an Agilent Biotek Cytation 1 Plate Reader using a Cy5 filter set (ex: 620 nm, em: 680 nm).

### Bacterial lysate production and norspermidine calibration

To produce bacterial lysates, cells from overnight 5-mL cultures grown in M9 media were collected by centrifugation at 8000×g for 5 min. Pelleted cells were resuspended in 1 mL of M9 medium. Resuspended cells were then subjected either to 10× freeze−thaw cycles in liquid nitrogen or sonication, in either case resulting in near-total lysis. The efficiency of lysis was quantified by measuring colony forming units (CFUs) of the overnight culture before and after lysis by tenfold serial dilutions on LB plates (Supplementary Fig. 4c). After lysis, cell debris was removed by centrifugation at 8000×g for 5 min, and the resulting supernatant was filtered through a 0.45-μm filter.

To quantify the concentration of lysate necessary to achieve a biofilm response in *V. cholerae* (Supplementary Fig. 4), *V. cholerae* cells were grown in a range of lysate concentrations, from 0.625 to 20% of the final growth medium composition, and biofilm biomass was quantified from timelapse microscopy images as described above. Under the same conditions *PvpsL-lux* output was monitored. To determine the concentration of norspermidine in lysate, we calibrated the *PvpsL-lux* response to wild-type lysate to known concentrations of norspermidine spiked into $\Delta nspC$ lysate (which lacks endogenous norspermidine), revealing that lysis of saturated wild-type *V. cholerae* cultures yielded ~30–50 μM norspermidine.

### Type-VI competition assay

*V. cholerae* was subjected to Type-VI killing by *A. baylyi* using a previously established protocol[13]. Briefly, overnight *V. cholerae* cultures expressing *mScarlet* under the control of the constitutive *tac* promoter along with a spectinomycin resistance cassette were diluted 1:1000 in LB and grown for 3 h at 37 °C to establish mid-log growth. Cultures were centrifuged at 8000× g for 5 min and subsequently resuspended in LB to a final concentration of $OD_{600} = 10$. Concentrated *V. cholerae* cultures were then mixed in a 1:1 ratio with *A. baylyi* and 5 μL of the mixture was spotted on an LB agar plate. After a 2-h incubation at 37 °C, mixtures were resuspended in 1 mL LB, serial dilutions were spotted on selective (spectinomycin-containing) LB agar plates, and after 18 h of growth at 30 °C, *V. cholerae* CFUs were measured using a FluoroChem M (ProteinSimple) imaging system. When applied, norspermidine was included at all stages of growth (overnight, during regrowth, and during the competition). Three biological replicates were performed.

### Phylogenetic tree construction

Core gene phylogenetic tree construction (Fig. 4A) was performed by downloading complete genomes for the relevant bacterial species from NCBI. A subset of 30 of these genomes were then selected for further analysis based on a sequence diversity maximization algorithm[46]. Core proteins with an amino acid identity cutoff of 30% were extracted with usearch[47] and subsequently aligned using the MUSCLE algorithm in the BGPA pipeline[48]. The alignment of core proteins was used to construct a maximum-likelihood phylogenetic tree with RAxML[49] via a WAG substitution model and gamma rate heterogeneity. Trees were visualized using iTOL, and species branches were colored according to the presence of a homolog of DABA AT/DC (the gene product of Vc_1625), which is the unique enzyme responsible for norspermidine biosynthesis via an established pathway or the presence of a homolog of MbaA (the gene product of Vc_0703) in any of the species' genomes. To identify homologs, a homology search for each of two gene product sequences (Vc_0703 or Vc_1625) was performed using BLAST with the non-redundant (nr) protein sequences database. A percent identity of ≥60% over ≥90% of the length of the gene product was used as the criterion for identifying homologs. Scripts were written in Python 3 and Bash.

### Reporting summary

Further information on research design is available in the Nature Portfolio Reporting Summary linked to this article.

## Data availability

The source data used to generate all main and Supplementary Figs. in this work are available on Figshare (https://doi.org/10.6084/m9.figshare.24488332). Accession codes for the genomes used to generate the species tree (Fig. 4a) are reported in Supplementary Table 5.

## Code availability

All custom processing scripts are available on Github (https://github.com/BridgesLabCMU/Lysis-sensing-scripts)[50].

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

## Acknowledgements

The authors thank members of the Bridges lab for insightful discussions and for reading the manuscript. We thank Dr. Thomas J. Silhavy for supplying *E. coli* BW25113 and the *speE*::*kanR* mutant, Dr. Courtney K. Ellison for supplying *A. baylyi* ADP1, and Dr. N.L. Hiller for supplying *P. aeruginosa* PA14. This work was supported by NIH grant R00AI158939, a Damon Runyon Cancer Research Foundation Dale F. Frey Award for Breakthrough Scientists, and startup funds from Carnegie Mellon University. The funders had no role in study design, data collection and analysis, decision to publish, or preparation of the manuscript.

## Author contributions

J.A.P. and A.A.B. were responsible for conceptualization, experimentation, data curation, investigation, methodology, validation, visualization, and writing the manuscript. J.A.P. was responsible for formal analysis and software. A.A.B. was responsible for funding acquisition, project administration, and supervision. A.A.B. and R.W. were responsible for resources. All authors reviewed and edited the manuscript.

## Competing interests

The authors declare no competing interests.
