## [Peer Review File · Nature Communications]

Cell-lysis sensing drives biofilm formation in *Vibrio cholerae*Editorial Note: Parts of this Peer Review File have been redacted as indicated to maintain the confidentiality of unpublished data.

Reviewer #1 (Remarks to the Author):

Prentice et al showed that biofilms, which are matrix-encapsulated bacterial communities, are common in the environment and challenging to manage in clinical and industrial contexts. They likely evolved as a protective mechanism against environmental threats. The process by which bacteria assess threats to inform biofilm development is not well understood. However, it has been found that cell lysis events at the population level prompt surviving *Vibrio cholerae* cells to form biofilms. These survivors detect threats by sensing a cellular component, norspermidine, released during cell lysis. This lysis sensing happens via the MbaA receptor with genus-level specificity, and the resulting biofilm cells are protected from phage infections and attacks from other bacteria. This research reveals a key link between bacterial lysis and biofilm formation, which could be widely applicable to other microorganisms.

The study was conducted meticulously and yielded clear, reliable results. The methodology was sound and the data was presented in a well-organized manner. The interpretation of the results was accurate and insightful. Overall, the research was executed to a high standard, demonstrating a strong understanding of the underlying principles.

Major concerns:

1) In Bridges' 2021 *eLife* paper (Inverse regulation of *Vibrio cholerae* biofilm dispersal by polyamine signals), he wrote "Spermidine, a polyamine frequently produced, facilitates the dispersal of *V. cholerae*, while norspermidine, a less common polyamine generated by vibrios, acts to prevent dispersal. The difference between spermidine and norspermidine is a single methylene group. Both of these polyamines regulate dispersal through the detection of MbaA in the periplasm and the following signal transmission." Therefore, the novelty of this finding can be considered low if they do not significantly advance our understanding of how norspermidine induces biofilm formation mechanistically or if they closely resemble existing biofilm knowledge. However, it's important to note that while such findings might not be groundbreaking, they still contribute to the broader scientific discourse and can serve as stepping stones for future phage-biofilm research. Similar studies listed below.

Wotanis, Caitlin K., et al. "Relative contributions of norspermidine synthesis and signaling pathways to the regulation of *Vibrio cholerae* biofilm formation." *PLoS One* 12.10 (2017): e0186291.

Karatan, Ece, Tammi R. Duncan, and Paula I. Watnick. "NspS, a predicted polyamine sensor, mediates activation of *Vibrio cholerae* biofilm formation by norspermidine." *Journal of bacteriology* 187.21 (2005): 7434-7443.

Parker, Zachary M., et al. "Elevated levels of the norspermidine synthesis enzyme NspC enhance *Vibrio cholerae* biofilm formation without affecting intracellular norspermidine concentrations." *FEMS microbiology letters* 329.1 (2012): 18-27.

2) Have you ever thought about the idea that eDNA and cellular debris, resulting from lysis initiated by lytic phage infection, could suggest that lysis mediated by phages in the initial stages of biofilm formation might have a positive effect on the formation of later structures? An interesting experiment could be to identify the receptor of phage S5 and investigate whether the deletion of nspC or mbaA influences the expression of the phage receptor, and how a phage receptor deletion mutant reacts to the addition of norspermidine.

3) The challenge lies in differentiating between norspermidine and quorum sensing (QS) mediated phage protection, given that QS-autoinducers are also released during cell lysis. If the deletion of nspC and mbaA significantly impacts QS, making it more prominent regulator during phage-host interactions, it aligns with numerous studies showing that QS suppresses the expression of phage receptors in *Vibrio* spp., such as O-antigen and OmpK. For example, a study by Hoque et al. found that a form of gene expression in bacteria, known as "quorum sensing," which is regulated by signal molecules known as autoinducers (AIs), can protect *V. cholerae* from predatory phages. *V. cholerae* mutant strains with deactivated AI synthase genes were significantly more susceptible to multiple phages than the parent bacteria. Similarly, when exogenous autoinducers CAI-1 or AI-2, produced by recombinant strains with cloned AI synthase genes, were added to mixed cultures of phage and bacteria, an increase in *V. cholerae* survival and a decrease in phage titer were observed. Mutational studies suggest that the effects of autoinducers are partly mediated through the quorum sensing-dependent production of haemagglutinin protease and partly through the downregulation of phage receptors.

Mimi questions:

Line 348 filtration through a 0.45 μM syringe filter. It's frequently observed that 0.45 μM is not always sufficient to completely sterilize phage lysate to plate phage lysate or treat it with chloroform if it shows resistance.

Line 354. What is the reason for selecting such a low Multiplicity of Infection (MOI), and what outcomes could be expected if a higher MOI is utilized? Given previous study has shown that low MOIs induce biofilm formation.

Fernández, Lucía, et al. "Low-level predation by lytic phage phiIPLA-RODI promotes biofilm formation and triggers the stringent response in *Staphylococcus aureus*." *Scientific reports* 7.1 (2017): 40965.

Line 381 Have you ever considered buying synthetic norspermidine and examining its impact on phage-host interactions and biofilm formation?

Reviewer #2 (Remarks to the Author):

This manuscript by Prentice, van de Weerd and Bridges reports on a series of quite straightforward experiments that convincingly demonstrates biofilm formation in response to kin lysis sensing in *Vibrio cholerae* and related *Vibrios*. Furthermore, the authors identify the lysis signal for the tested strains of *V. cholerae* and *V. anguillarum* as norspermidine, a polyamine synthesized by a limited subset of *Vibrio* bacteria. In a sense, the manuscript is the culmination of previous work by the last author, demonstrating biofilm formation in response to norspermidine, and identification of the signaling pathway (ref 20-22 in manuscript). In the current manuscript, we now learn why it would benefit *V. cholerae* to upregulate stable biofilm formation in response to norspermidine, namely because the presence of extracellular norspermidine indicates lysis of kin bacteria in the surroundings, and thus indicates a threat to the survival of planctonic cells, such as phage infection, which can be mitigated by sustained biofilm formation.

I believe this work will greatly impact the microbiology community, especially the many researchers who study phage-bacterial interactions and interbacterial "warfare", because until now it has been unknown how an individual bacterium might sense an incoming threat, prior to actually experiencing the threat (e.g. phage infection).

I consider the experimental work of high quality and have just a few comments for the authors:

-line 59: From Figure 1, I agree it is evident that the bacteria who survive phage S5 are in the biofilm state. But couldn't it be that the phage simply selects for bacteria in the biofilm state by killing off the planctonic cells? In other words, how do the authors believe fig. 1 confirms "that the presence of phage drives biofilm gene expression", rather than that the presence of phage selects for growth of a subpopulation that was already expressing biofilm genes?

-the authors should make their definition of biofilm more clear. In line 133-34 it is stated that the mutants form "multicellular aggregate-like structures" in the presence of phage. That sounds like biofilm. I understand that this biofilm is not VpsL-driven, but that doesn't mean it is not biofilm. The authors should consider referring to all the aggregates as biofilm, and the subtype of biofilm they study as "vps-dependent biofilm" or similar.

-Phage-driven biofilm formation was previously reported in *Vibrio anguillarum*. I don't think this fact detracts from the current work, as the previous work does not identify the sensing mechanism at all, but the authors should cite the relevant literature, primarily the work by Tan, Dahl & Middelboe, 2015: *Vibriophages differentially influence biofilm formation*. doi: 10.1128/AEM.00518-15.

- line 176: I disagree with the statement that phage specificity is often at the genus or species-level. Many vibriophages show much more narrow specificity, down to the level of specific

strains/isolates. Similarly, strain-specific variation of the regulation of biofilm formation is observed in *Vibrio anguillarum*. It is therefore important to emphasize in the manuscript that these responses were observed for the particular strain of *V. cholerae*, *V. anguillarum*, etc that were tested, but may not broadly apply at the species level.

Reviewer #3 (Remarks to the Author):

Prentice et al. described that *V. cholerae* sense norspermidine released from lysed cells, which enhances biofilm formation. Biofilm formation in response to norspermidine has a protective effect from S5 phage infection. The previous works containing the author's papers already found norspermidine-mediated biofilm induction; the mechanism is already known. However, this study promotes the idea of the connection between norspermidine signaling and bacterial threat recognition and following community formation. Also, this study found that genes required for norspermidine production and sensing are conserved in the kin of Vibrionaceae and specific to limited *Vibrio* members. Data represent that sensing norspermidine from lysed cells could be a mechanism to sense and respond to dangerous environments for kin bacteria. Collectively, this concept will intrigue many readers, and the story presented is well-written, straightforward, and consistent. However, I have some concerns listed below.

The authors employed weak MOI conditions (10^{-6}) for the experiments. The reviewer is curious about the extent to which lysis-sensing biofilm formation confers tolerance to phage infection. Higher MOI conditions should be tried to emphasize the authors' claim that lysis-sensing contributes to the survival of phage threats. Also, the burst size of the S5 phage is 10, which is lower than that of other vibriophages. Does the lysis sensing system work for other phage infections whose burst size is more extensive?

Related to the above comment, the authors claim lysis sensing could benefit other lytic agents. Fig. S4 shows no difference in tolerance to *A. baylyi* attack between wt and *mbaA* mutant. If the lysis sensing system has protective effects on other bacterial attacks by T6SS, a subpopulation of wt cells are lysed by T6SS, and induces robust biofilms, thus protecting them from T6SS compared to *mbaA* mutant. How do the authors explain this discrepancy? Or is prior biofilm induction necessary for protection from the attack? It might be possible that the authors' claim that the lysis sensing system contributes to survival may not necessarily hold for all cases. In addition, in the context of treatment, the relevance of antibiotic tolerance is intriguing. Did the authors test if the *mbaA*-mediated system is involved in collective antibiotic tolerance?

P5, L100. It needs to be clarified whether norspermidine is sufficient or not. Also in the case of *V. anguillarum*, only norspermidine does not show maximal induction of biofilm formation. Is it possible that both norspermidine and other cellular components are needed for lysis-dependent biofilm induction? The authors should clarify that point.

In Fig. 1c and confocal images in Fig. 2 and Fig. 5, the authors should show the results of the control experiment.

Below are our point-by-point responses to the reviewers' comments. Reviewer comments are in black text and our responses are in green text. References to line numbers in the revised manuscript are relevant when Track Changes are set to "no markup."

Yours,

Drew Bridges

REVIEWER COMMENTS

Reviewer #1 (Remarks to the Author):

Prentice et al showed that biofilms, which are matrix-encapsulated bacterial communities, are common in the environment and challenging to manage in clinical and industrial contexts. They likely evolved as a protective mechanism against environmental threats. The process by which bacteria assess threats to inform biofilm development is not well understood. However, it has been found that cell lysis events at the population level prompt surviving *Vibrio cholerae* cells to form biofilms. These survivors detect threats by sensing a cellular component, norspermidine, released during cell lysis. This lysis sensing happens via the MbaA receptor with genus-level specificity, and the resulting biofilm cells are protected from phage infections and attacks from other bacteria. This research reveals a key link between bacterial lysis and biofilm formation, which could be widely applicable to other microorganisms. The study was conducted meticulously and yielded clear, reliable results. The methodology was sound and the data was presented in a well-organized manner. The interpretation of the results was accurate and insightful. Overall, the research was executed to a high standard, demonstrating a strong understanding of the underlying principles.

We thank the reviewer for their positive comments and insightful suggestions.

Major concerns:

1) In Bridges' 2021 *elife* paper (Inverse regulation of *Vibrio cholerae* biofilm dispersal by polyamine signals), he wrote "Spermidine, a polyamine frequently produced, facilitates the dispersal of *V. cholerae*, while norspermidine, a less common polyamine generated by vibrios, acts to prevent dispersal. The difference between spermidine and norspermidine is a single methylene group. Both of these polyamines regulate dispersal through the detection of MbaA in the periplasm and the following signal transmission." Therefore, the novelty of this finding can be considered low if they do not significantly advance our understanding of how norspermidine induces biofilm formation mechanistically or if they closely resemble existing biofilm knowledge. However, it's important to note that while such findings might not be groundbreaking, they still contribute to the broader scientific discourse and can serve as stepping stones for future phage-biofilm research. Similar studies listed below.

Wotanis, Caitlin K., et al. "Relative contributions of norspermidine synthesis and signaling pathways to the regulation of *Vibrio cholerae* biofilm formation." *PLoS One* 12.10 (2017): e0186291.

Karatan, Ece, Tammi R. Duncan, and Paula I. Watnick. "NspS, a predicted polyamine sensor, mediates activation of *Vibrio cholerae* biofilm formation by norspermidine." *Journal of bacteriology* 187.21 (2005): 7434-7443.

Parker, Zachary M., et al. "Elevated levels of the norspermidine synthesis enzyme NspC enhance *Vibrio cholerae* biofilm formation without affecting intracellular norspermidine concentrations." *FEMS microbiology letters* 329.1 (2012): 18-27.

We disagree with the reviewer's assessment of novelty. The reviewer is correct that we and others have previously characterized molecular mechanisms by which norspermidine drives biofilm formation in *V. cholerae*. The major advance of the current work (also pointed out by Reviewers #2 and #3), is in providing a functional and ecological link between norspermidine as a lysis-sensing signal and resulting biofilm formation for protection from threats. Collectively, this work reveals a novel mechanism (lysis sensing) by which bacterial cells sense an incoming threat and preemptively protect themselves via biofilm formation that is not limited to the pathway discussed here. To this end, we provide evidence that the principle of lysis sensing occurs in other *Vibrios* via yet-to-be discovered signal transduction pathways, and we speculate that it likely occurs in other diverse microbial systems. Thus, we firmly believe the discoveries in this manuscript will be of broad interest to researchers in diverse fields.

2) Have you ever thought about the idea that eDNA and cellular debris, resulting from lysis initiated by lytic phage infection, could suggest that lysis mediated by phages in the initial stages of biofilm formation might have a positive effect on the formation of later structures? An interesting experiment could be to identify the receptor of phage S5 and investigate whether the deletion of *nspC* or *mbaA* influences the expression of the phage receptor, and how a phage receptor deletion mutant reacts to the addition of norspermidine.

We thank the reviewer for suggesting this astute hypothesis. Indeed, it is entirely possible that in the context of a lysis event, biofilm architecture could be influenced by the presence of cellular polymers (e.g. eDNA) released into the surrounding medium, for instance, via a depletion mechanism, as has been described previously (reference 26 in the revised manuscript). In the case of the lysis sensing mechanism described in this work, our genetic evidence presented in Figs. 2-3 demonstrates that the WT biofilm response to lysate is exclusively mediated by signaling via norspermidine and subsequent upregulation of *vps*, and not an unidentified structural component. That said, such a mechanism may explain the alternative aggregate-type structures that form in the $\Delta vpsL$ and norspermidine-signaling mutants in response to phage infection (Fig. S5). To this end, we have included a sentence in the revised manuscript addressing the involvement of such a mechanism in this context:

Line 153: We speculate that the aggregate biofilms observed in the aforementioned mutant strains could result from cellular polymers released by lysis that induce

*aggregation by a depletion mechanism, as previously observed for other bacterial species.*²⁶

Finally, we agree with the reviewer that, going forward, it will be of great importance to identify the phage S5 receptor, and to characterize potential regulatory connections to NspS/MbaA. That said, we believe this work is beyond the scope of the current manuscript, as was also pointed out by the editor.

3) The challenge lies in differentiating between norspermidine and quorum sensing (QS) mediated phage protection, given that QS-autoinducers are also released during cell lysis. If the deletion of *nspC* and *mbaA* significantly impacts QS, making it more prominent regulator during phage-host interactions, it aligns with numerous studies showing that QS suppresses the expression of phage receptors in *Vibrio* spp., such as O-antigen and OmpK. For example, a study by Hoque et al. found that a form of gene expression in bacteria, known as “quorum sensing,” which is regulated by signal molecules known as autoinducers (AIs), can protect *V. cholerae* from predatory phages. *V. cholerae* mutant strains with deactivated AI synthase genes were significantly more susceptible to multiple phages than the parent bacteria. Similarly, when exogenous autoinducers CAI-1 or AI-2, produced by recombinant strains with cloned AI synthase genes, were added to mixed cultures of phage and bacteria, an increase in *V. cholerae* survival and a decrease in phage titer were observed. Mutational studies suggest that the effects of autoinducers are partly mediated through the quorum sensing-dependent production of haemagglutinin protease and partly through the downregulation of phage receptors.

We thank the reviewer for pointing out the connection between QS and protection from phage. We believe this is an interesting point that we now discuss in the revised manuscript (see below). While we agree that autoinducers could, in principle, be released from lysed cells, this mechanism alone is not sufficient to explain the protection mediated by lysis sensing as described here. Firstly, we find that lysis-sensing protection is mediated by biofilm formation, a phenotype that is generally repressed by quorum sensing autoinducers in *V. cholerae* (Hammer and Bassler, 2003. DOI: 10.1046/j.1365-2958.2003.03688.x). Moreover, our genetic evidence (using *nspC* and *mbaA* mutants) demonstrates that biofilm-mediated protection by norspermidine signaling is necessary and sufficient to explain our observations.

That said, previous work by members of our laboratory explored how quorum sensing and norspermidine signaling are interconnected. We found that deletion of either *nspC* or *mbaA* has no effect on QS gene expression (citation 23 in the revised manuscript). On the other hand, we also found that the high-cell density QS state enhances the effect of norspermidine on biofilm formation by increasing MbaA levels (citation 36 in the revised manuscript), indicating that QS may sensitize populations to the lysis sensing mechanism described here. In the Discussion section of the revised manuscript, we now include the following statement:

*Line 294: Bacteria often encounter multiple environmental inputs simultaneously that can either synergize or interfere with one another via signal integration networks. Investigating how cells integrate lysis sensing with other signaling inputs could reveal the ecological settings in which lysis sensing is most relevant. Previous results indicated a relationship between the lysis sensing receptor, MbaA, and quorum sensing. Quorum sensing is the process by which bacteria secrete and detect small molecule signals, allowing them to measure the cell density and relatedness of their community.³⁴ Bacteria including *V. cholerae* have been shown to upregulate other phage protection mechanisms at high cell densities via quorum sensing, presumably to protect themselves when the population is most vulnerable to phage propagation, i.e., when cell numbers are large.³⁵ We previously showed that quorum sensing, which under normal conditions represses biofilm formation in *V. cholerae*, amplifies biofilm formation in the presence of the norspermidine signal via upregulation of the MbaA receptor.³⁶ Thus, *V. cholerae* cells are more sensitive to lysis signals when the potential for phage propagation is highest. Based on these findings, and their functional logic, we speculate that a regulatory link between lysis sensing and quorum sensing may be common in bacteria.*

Mimi questions:

Line 348 filtration through a 0.45 µM syringe filter. It's frequently observed that 0.45 uM is not always sufficient to completely sterilize phage lysate to plate phage lysate or treat it with chloroform if it shows resistance.

We thank the reviewer for pointing this out. We have repeated our phage killing assay using phage filtered through a 0.2 µm syringe filter (Fig. S1B) and have observed no difference in either the killing curve or the biofilm phenotype. We have included a line in the Methods stating:

Line 409: Filtration of phage preparations through 0.2 µm or 0.45 µm syringe filters resulted in similar phage preparations.

We additionally plated the phage lysate preparation that had been filtered through a 0.45 µm syringe filter and observed no contamination (data not shown).

Line 354. What is the reason for selecting such a low Multiplicity of Infection (MOI), and what outcomes could be expected if a higher MOI is utilized? Given previous study has shown that low MOIs induce biofilm formation.

Fernández, Lucía, et al. "Low-level predation by lytic phage phiPLA-RODI promotes biofilm formation and triggers the stringent response in *Staphylococcus aureus*." Scientific reports 7.1 (2017): 40965.

We thank the reviewer for this suggestion. We have performed the same experiment as in Fig. 1B at a range of higher MOI's (Fig. S1B of the revised manuscript). Our modifications to the text are as follows:

Line 56: Both biofilm formation and growth recovery were observed at a range of MOIs (up to MOI ~0.1) in response to infection (Fig. S1B).

Line 381 Have you ever considered buying synthetic norspermidine and examining its impact on phage-host interactions and biofilm formation?

Indeed, we used synthetic norspermidine for numerous experiments in this manuscript (e.g., Fig. 2 and Fig. 4). These experiments showed that exogenous norspermidine is sufficient to reconstitute the biofilm response to lysate.

Reviewer #2 (Remarks to the Author):

This manuscript by Prentice, van de Weerd and Bridges reports on a series of quite straightforward experiments that convincingly demonstrates biofilm formation in response to kin lysis sensing in *Vibrio cholerae* and related *Vibrios*. Furthermore, the authors identify the lysis signal for the tested strains of *V. cholerae* and *V. anguillarum* as norspermidine, a polyamine synthesized by a limited subset of *Vibrio* bacteria. In a sense, the manuscript is the culmination of previous work by the last author, demonstrating biofilm formation in response to norspermidine, and identification of the signaling pathway (ref 20-22 in manuscript). In the current manuscript, we now learn why it would benefit *V. cholerae* to upregulate stable biofilm formation in response to norspermidine, namely because the presence of extracellular norspermidine indicates lysis of kin bacteria in the surroundings, and thus indicates a threat to the survival of planctonic cells, such as phage infection, which can be mitigated by sustained biofilm formation.

I believe this work will greatly impact the microbiology community, especially the many researchers who study phage-bacterial interactions and interbacterial "warfare", because until now it has been unknown how an individual bacterium might sense an incoming threat, prior to actually experiencing the threat (e.g. phage infection).

I consider the experimental work of high quality and have just a few comments for the authors:

We thank the reviewer for their complimentary feedback and valuable suggestions.

-line 59: From Figure 1, I agree it is evident that the bacteria who survive phage S5 are in the biofilm state. But couldn't it be that the phage simply selects for bacteria in the biofilm state by killing off the planctonic cells? In other words, how do the authors believe fig. 1 confirms "that the presence of phage drives biofilm gene expression", rather than that the presence of phage selects for growth of a subpopulation that was already expressing biofilm genes?

We thank the reviewer for this comment. If a subpopulation of *V. cholerae* cells were intrinsically expressing biofilm genes, as the reviewer suggests, we would expect to see that subpopulation forming biofilms in the absence of phage (i.e., some cells in our large

fields of view would form biofilms). However, we observed no biofilm formation in the absence of phage (Fig. 1A; see below for images of the fields of view).

Moreover, our reconstitution of the biofilm phenotype with crude lysates (no phage), the genetic evidence implicating NspC/MbaA, and the observed fitness differences between the WT and *nspC/mbaA* mutants in the presence of phage (Figs. 2-3) together demonstrate that lysis sensing is the dominant mechanism underpinning our observations. That said, we do agree with the reviewer that we should acknowledge the potential for a mechanism by which phage selects for cells expressing elevated biofilm genes. To this end we have modified the text accordingly:

Line 77: We considered three possibilities: 1) phage killing selects for cells that exhibit elevated vps expression, 2) V. cholerae cells sense phage components during infection and form biofilms in response, or 3) lysis releases a cytoplasmic signal, and living cells respond to this cue by committing to the biofilm state. We reasoned that one could distinguish between these possibilities by exposing cells to mechanically produced cell lysate lacking any phage component....

*Line 140: To validate our findings, we evaluated biofilm gene expression using P_{vpsL}-lux. Consistent with our biofilm measurements, P_{vpsL}-lux output from the wildtype strain was ~7-fold greater than that of the Δ *nspC* mutant and was ~24-fold greater than that of the *mbaA** mutant in the presence of phage (Fig. 3B). These results demonstrate that lysis sensing, and not selection of cells exhibiting elevated vps gene expression, accounts for the elevation in biofilm formation in the presence of phage.*

-the authors should make their definition of biofilm more clear. In line 133-34 it is stated that the mutants form "multicellular aggregate-like structures" in the presence of phage. That sounds like biofilm. I understand that this biofilm is not VpsL-driven, but that doesn't mean it is not biofilm. The authors should consider referring to all the aggregates as biofilm, and the subtype of biofilm they study as "vps-dependent biofilm" or similar.

We thank the reviewer for this suggestion. In the revised manuscript, we have clarified our terminology exactly as suggested in the relevant locations:

Line 147: We note that the $\Delta nspC$ and $mbaA^$ mutants did appear to form aggregate-like biofilms in the presence of phage that were not observed in the WT strain and, due to their distinct optical properties, were not segmented by our image analysis pipeline developed for detecting canonical VPS-dependent biofilms (Fig. S5, Movie S3). Of note, these aggregate biofilms were also observed in a $\Delta vpsL$ strain, which lacks a critical polysaccharide matrix biosynthesis enzyme required for canonical VPS-dependent biofilm formation (Fig. S5).*

Fig. S5 legend: The black arrow in the WT image indicates VPS-dependent biofilms, which are segmented in our image analysis pipeline. Red arrows indicate more diffuse aggregate-like biofilms (red) which form in the mutants and do not meet the threshold conditions in our image analysis pipeline.

-Phage-driven biofilm formation was previously reported in *Vibrio anguillarum*. I don't think this fact detracts from the current work, as the previous work does not identify the sensing mechanism at all, but the authors should cite the relevant literature, primarily the work by Tan, Dahl & Middelboe, 2015: *Vibriophages differentially influence biofilm formation*. doi: 10.1128/AEM.00518-15.

We thank the reviewer for pointing out this oversight. now acknowledge this previous work as follows:

*Line 235: We note that a previous study observed that *V. anguillarum* formed biofilms in response to phage, but the mechanism underpinning the response was not identified.²⁹ Our results show that the norspermidine-dependent lysis sensing mechanism is functionally conserved in a subset of *Vibrio* species, and that lysis sensing could explain the previous observation of phage-induced biofilm formation in *V. anguillarum*.*

- line 176: I disagree with the statement that phage specificity is often at the genus or species-level. Many vibriophages show much more narrow specificity, down to the level of specific strains/isolates. Similarly, strain-specific variation of the regulation of biofilm formation is observed in *Vibrio anguillarum*. It is therefore important to emphasize in the manuscript that these responses were observed for the particular strain of *V. cholerae*, *V. anguillarum*, etc that were tested, but may not broadly apply at the species level.

We thank the reviewer for clarifying this point about specificity. We have changed our phrasing of the phage specificity point on the following line:

Line 196: Pervasive lytic threats (e.g., phage) often target prey with high specificity – down to the level of single strains or isolates, and thus, threat assessment is likely most relevant if information about the threat is encoded in a “kin”-specific lysis signal.

Additionally, in the Discussion section of the revised manuscript, we have added the caveat that our results may be specific to the particular *Vibrio* strains we tested, and that further work will be required to assess the generality of the biofilm response to lysis:

Line 269: Future work will be required determine whether lysis sensing control of biofilm formation extends beyond the particular species and strains employed in this study.

Reviewer #3 (Remarks to the Author):

Prentice et al. described that *V. cholerae* sense norspermidine released from lysed cells, which enhances biofilm formation. Biofilm formation in response to norspermidine has a protective effect from S5 phage infection. The previous works containing the author's papers already found norspermidine-mediated biofilm induction; the mechanism is already known. However, this study promotes the idea of the connection between norspermidine signaling and bacterial threat recognition and following community formation. Also, this study found that genes required for norspermidine production and sensing are conserved in the kin of Vibrionaceae and specific to limited *Vibrio* members. Data represent that sensing norspermidine from lysed cells could be a mechanism to sense and respond to dangerous environments for kin bacteria. Collectively, this concept will intrigue many readers, and the story presented is well-written, straightforward, and consistent. However, I have some concerns listed below.

We thank the reviewer for their positive view of our manuscript and helpful suggestions.

The authors employed weak MOI conditions (10^{-6}) for the experiments. The reviewer is curious about the extent to which lysis-sensing biofilm formation confers tolerance to phage infection. Higher MOI conditions should be tried to emphasize the authors' claim that lysis-sensing contributes to the survival of phage threats. Also, the burst size of the S5 phage is 10, which is lower than that of other vibriophages. Does the lysis sensing system work for other phage infections whose burst size is more extensive?

We thank the reviewer for this suggestion. We have performed the same experiment as in Fig. 1B at higher MOI's (results are below and in Fig. S1B of the revised manuscript). We continue to observe both a biofilm response and recovery at all tested MOI's (up to MOI ~ 0.1 – the highest MOI we could test with the titer of our phage stock and the initial concentration of bacteria), demonstrating that lysis sensing is functional even at high phage titers. The primary difference we observe is that at higher MOIs, the initial growth phase is attenuated. Moreover, regarding phage infections with larger burst sizes, we have acquired vibrio phage N4, for which we performed a one-step growth curve (new Fig S1C, right panel), revealing a burst size of ~50 particles/infection. We observe that *V. cholerae* exhibits a similar biofilm response in the presence of phage N4 (Fig. S1C). The modified text noting these experiments reads as follows:

Line 56: Both biofilm formation and growth recovery were observed at a range of MOIs (up to MOI ~0.1) in response to infection (Fig. S1B).

Line 62: Of note, when we treated V. cholerae with the lytic phage N4, which exhibits a larger burst size (~50 particles/cell compared to ~10 particles/cell for S5; Fig. S1C), we observed a similar increase in biofilm formation and elevated PvpsL-lux output relative to untreated cultures, showing that the biofilm response to lytic phage is not unique to a particular phage (Fig. S1C).¹⁷

Though not achieved with the phages used in this work, we agree with the reviewer's point that lysis sensing protection should be overwhelmed at exceedingly high phage concentrations. The noteworthy points raised by the reviewer both here and below, with regard to antibiotics, have led us to include a new discussion paragraph that focuses on the bounds of lysis sensing:

Line 283: The biophysical characteristics of threats and the properties of lysis sensing circuits (e.g., sensitivity) are likely critical for the effectiveness of lysis sensing in protecting bacterial populations. Regarding the properties of threats, we reason that the amount of the threatening agent, its diffusivity, and its propagation rate are crucial parameters. A population of bacteria presumably cannot protect themselves against a threat that diffuses rapidly and affects all members of a population simultaneously (e.g., antibiotics, or a very high concentration of phage) using lysis sensing, which depends on the comparatively longer timescales of lysate accumulation and the response to a lysis signaling molecule. On the other hand, threats that propagate through a population over time and diffuse slower than lysis signals, such as dilute phage and bacterial or protozoan predators, are likely candidates that could drive the evolution of lysis sensing.

Related to the above comment, the authors claim lysis sensing could benefit other lytic agents. Fig. S4 shows no difference in tolerance to A. baylyi attack between wt and mbaA mutant. If the lysis sensing system has protective effects on other bacterial attacks by T6SS, a subpopulation of wt cells are lysed by T6SS, and induces robust biofilms, thus protecting them from T6SS compared to mbaA mutant. How do the authors explain this discrepancy? Or is prior biofilm induction necessary for protection from the attack? It might be possible that the authors' claim that the lysis sensing system contributes to survival may not necessarily hold for all cases.

We thank the reviewer for noting this, and we agree with their points. We believe that the equivalent survival of the untreated WT and mbaA* populations in the T6SS assay is largely a function of the configuration of the T6SS killing assay. We have added the following lines to the revised text:

Line 179: Notably, pre-exposure to the lysis signal was required for a survival advantage as we observed no survival difference between the untreated WT and mbaA strains in this standard T6SS competition assay. We reason that due to the short timescale of incubation (2 hours), lysis sensing occurring concomitant with exposure to high number of A. baylyi cells does not provide enough time for biofilm-mediated protection to be enacted before V. cholerae is overwhelmed by attacks. These results suggest that the lysis signaling mechanism that allows cells to collectively respond to phage-mediated killing may allow them to respond similarly to other lytic threats.*

In addition, in the context of treatment, the relevance of antibiotic tolerance is intriguing. Did the authors test if the mbaA-mediated system is involved in collective antibiotic tolerance?

We thank the reviewer for suggesting this experiment.

[REDACTED]

REDACTED

We add that in principle, lysis sensing is less likely to be effective against antibiotics than it is against threats which propagate, like phage, or predators. This is because propagating threats do not affect all members of the population at once and therefore give the unaffected fraction time to respond to lysis of their kin. We cover this concept in the new discussion paragraph (Line 283 in the manuscript, copied above in response to MOI/burst size suggestion).

Given the lack of a clear mechanistic connection between antibiotic treatment and lysis sensing, we propose to withhold these results from the current manuscript.

P5, L100. It needs to be clarified whether norspermidine is sufficient or not. Also in the

case of *V. anguillarum*, only norspermidine does not show maximal induction of biofilm formation. Is it possible that both norspermidine and other cellular components are needed for lysis-dependent biofilm induction? The authors should clarify that point.

We thank the reviewer for raising this issue. Indeed, in Fig. 2B, C, our genetic results demonstrate that norspermidine is both necessary and sufficient for the biofilm response to lysate in *V. cholerae*:

*Line 102: ... we treated wildtype cells with lysate derived from a Δ nspC mutant, which does not encode the carboxynorspermidine decarboxylase required for norspermidine biosynthesis (Fig. S3).²⁴ In response to the norspermidine deficient lysate, wildtype cells did not exhibit elevated biofilm biomass relative to untreated cells (Fig. 2B, Movie S2). However, when we administered lysate from the Δ nspC strain spiked with exogenous norspermidine, we observed robust biofilm formation and attenuated dispersal, akin to the response to wildtype lysate (Fig. 2C). Thus, norspermidine is necessary and sufficient for the biofilm response to lysate in *V. cholerae*.*

For *V. anguillarum*, the reviewer is correct to point out that treatment with norspermidine does not enhance biofilm formation to the same extent as lysate. Indeed, it is possible that other cellular components contribute to lysis-dependent biofilm induction in *V. anguillarum*. In the revised manuscript, we have added an additional sentence clarifying this point:

*Line 232: Notably, however, *V. anguillarum* exhibited greater biofilm formation in the presence of its own lysate than in the presence of synthetic norspermidine (Fig. 5A), suggesting that other cellular components may synergize with norspermidine to drive lysis sensing in this bacterium.*

In Fig. 1c and confocal images in Fig. 2 and Fig. 5, the authors should show the results of the control experiment.

We thank the reviewer for pointing out this oversight. For each of the mentioned experiments, we have now included supplemental figures containing the controls (Fig. S1A, Fig. S2, and Fig. S8 in the revised manuscript).